# Security in Wireless Sensor Networks Using OMNET++: Literature Review

**DOI:** 10.3390/s25102972

**Published:** 2025-05-08

**Authors:** Maahiya Shaik, Sung Won Kim

**Affiliations:** 1Department of Information and Communication Engineering, Yeungnam University, Gyeongsan 38428, Republic of Korea; maahiyashaik2000@gmail.com; 2School of Computer Science and Engineering, Yeungnam University, Gyeongsan 38428, Republic of Korea

**Keywords:** OMNET++, intrusion detection

## Abstract

With the essential increase in the use of wireless sensor networks, security is a major concern in every field. Intrusions have become frequent and present a significant challenge in today’s world. It is valuable to explore the feasibility of designing and rigorously assessing intrusion detection systems within network simulation environments. Wireless sensor network security risk prediction is a key aspect of wireless network security technology. Analyzing the current state of wireless networks, security is a crucial step in ongoing research in the field of network security. In this paper, we discuss how OMNET++ is used for intrusion detection for different types of attacks in wireless sensor networks, what frameworks and protocols are used in OMNET++, and why OMNET++ is used, along with a few security attacks in wireless networks.

## 1. Introduction

Wireless sensor networks (WSNs) have emerged as a groundbreaking innovation in contemporary communication and data gathering systems. Comprising numerous independent sensor nodes, these networks track a range of physical and environmental parameters, including temperature, humidity, movement, and sound, and collaboratively transmit the collected information to a centralized hub. Their adaptability, scalability, and resilience in challenging or remote settings have positioned WSNs as vital tools across diverse sectors such as environmental conservation, medical technology, defense operations, agriculture, and urban development.

WSNs encompass a broad array of wireless communication technologies that eliminate the need for physical wiring. Leveraging protocols such as Wi-Fi, Bluetooth, satellite links, and cellular connectivity like 4G and 5G, these systems have become foundational in modern data-driven applications. Their adaptability and ease of deployment allow them to be used across diverse sectors, from monitoring environmental changes and automating industrial operations to enhancing healthcare systems, securing military zones, and powering smart infrastructure. Their capability to operate in remote, hazardous, or previously unreachable locations makes them a highly valuable asset in the digital era. However, with these advantages comes a set of significant hurdles, particularly in securing wireless communications. Unlike traditional wired setups, where data travel along protected physical lines, WSNs rely on open-air transmission mediums such as radio frequencies. This inherent exposure makes them susceptible to various security risks, including data transmission [1,2], unauthorized modification, and impersonation attacks. Without effective safeguards, malicious actors can easily exploit these vulnerabilities, compromising not just individual nodes but entire networks. Consequently, achieving secure transmission ensuring data remain private, accurate, and authentic has emerged as a pressing challenge in WSN research and deployment.

The communication process in WSNs typically involves transmitting information from sensor nodes to a centralized unit or base station, often traversing multiple intermediary nodes. These data packets may contain critical insights, from real-time medical diagnostics to sensitive defense-related surveillance, underscoring the need for robust protection. However, the limited processing power, storage, and energy reserves of most sensor nodes restrict the implementation of complex cryptographic techniques. As a result, current research efforts are intensively focused on developing lightweight, energy-conscious security solutions that safeguard wireless transmissions while preserving network longevity and performance. These networks are crucial for mobile communication, Wi-Fi, satellite systems, and many other applications. However, wireless data transmission faces several challenges, including interference, signal degradation over distance (attenuation), security vulnerabilities, limited bandwidth, and issues related to multipath propagation.

There are many layer-specific attacks in wireless networks, and the five key layers are shown in Figure 1: the physical layer, MAC layer, network layer, transport layer, and application layer. The physical layer, as the foundational layer of WSNs, handles frequency band management and ensures data integrity. However, it is highly susceptible to interference and direct attacks on nodes, making these attacks particularly challenging to prevent. The MAC layer is responsible for error detection and monitoring radio channel usage [3]. Attackers can intercept entire data packets during transmission, forcing nodes to retransmit and rapidly depleting their energy. Countermeasures include using smaller packet sizes and advanced encryption techniques. The network layer facilitates routing between source and destination nodes and is prone to multiple attack types, such as wormhole, black hole, DoS, Sybil, HELLO flood, and selective forwarding attacks [4]. The transport layer plays a vital role in ensuring data delivery accuracy and controlling traffic overload within the network. It incorporates specialized protocols designed for wireless sensor environments, each addressing reliability from a different angle. Common threats to this layer include flooding attacks and resource depletion. The application layer focuses on configuring the network for specific purposes, such as extracting and managing WSN topologies. It oversees traffic management and data processing, but vulnerabilities here can result in inaccurate or unreliable data outputs.

While wireless sensor networks (WSNs) offer tremendous advantages, they are equally vulnerable to a range of security threats because of their inherent limitations, namely restricted battery life, processing capabilities, and memory. In wireless networks, there are small, low-cost, resource-constrained nodes, referred to as sensors, which possess the ability to sense, process, and communicate data. However, these nodes are constrained by factors such as limited computational power, finite energy resources, and restricted memory capacity. Critical security concerns include the following: resource limitations, data interception through eavesdropping, denial-of-service (DoS) attacks such as synchronize flood attacks [5], DDoS attacks [6], replication, sinkhole attacks [7], and selective forwarding [8], which disrupt normal operations, routing disruptions, and challenges related to authentication and trust. Conventional security frameworks, which often rely on complex encryption methods, are unsuitable for WSNs as the sensor nodes cannot handle heavy computational loads. Consequently, the development of lightweight, energy-efficient protection mechanisms is vital. The wireless nature of WSNs exposes communications to interception risks, endangering sensitive data such as healthcare records or defense information. Moreover, the physical compromise of sensor nodes in unsecured or hostile environments can allow adversaries to extract encryption keys or reprogram devices for malicious purposes. Networks can also be overwhelmed through illegitimate traffic floods [9], depleting node energy and severely affecting communication, a potentially catastrophic event in critical systems. Since WSNs often depend on multi-hop routing, they are susceptible to attacks like sinkholes, wormholes, and selective forwarding, which can disrupt data flow or partition the network. Additionally, maintaining secure node authentication and building node trust [10] among devices in expansive WSN deployments is a daunting yet necessary task to prevent internal threats.

The contributions of this paper are as follows: In this paper, an overview of wireless sensor networks (WSNs) is presented, outlining their core concepts and highlighting the primary security challenges they face. We also delve into the diverse application areas of WSNs, including domains like cybersecurity and data safeguarding. A significant focus of this work is on illustrating the foundational techniques and methodologies applied through OMNET++ in WSN simulations.

The structure of this paper is as follows:An analysis of the strengths and weaknesses of OMNET++ when applied to wireless sensor networks, emphasizing its role in enhancing network security and the hurdles encountered.A review of the current literature related to OMNET++ simulations using wireless sensor nodes, aiming to shed light on how OMNET++ has been employed across different wireless network environments.A demonstration of various case studies, highlighting OMNET++’s advantages, while also examining the frameworks and architectures integrated within OMNET++ simulations.

The next sections are organized as follows: In Section 2, a literature review of the existing work related to OMNET++ is given. In Section 3, OMNET++ and the basic concepts, model, and frameworks of OMNET++ are discussed. In Section 4, the importance of OMNET++ and its uniqueness are elaborated. In Section 5, security in wireless networks is shown, along with a few attack scenarios in wireless networks using OMNET++.

## 2. Literature Review

This section presents a comprehensive review of prior studies and associated research within the field. The main goal is to summarize the current state of knowledge, identify areas where further investigation is needed, and critically assess the shortcomings and constraints of existing methods.

### 2.1. Foregoing Work

In the current landscape of wireless sensor network (WSN) security research, a significant emphasis has been placed on the foundational objectives of protecting data and communication, particularly focusing on confidentiality, data integrity, secure authentication, and ensuring the continuous availability of services. Initial research efforts were geared toward developing resource-efficient solutions, recognizing the limitations in processing power, memory, and energy inherent to sensor nodes. These efforts included the design of lightweight encryption techniques [11] that minimize overhead, the implementation of secure routing protocols [12] to prevent unauthorized path manipulation, and the integration of intrusion detection mechanisms [9] capable of identifying irregular behavior with minimal energy consumption. Trust-based models [13] were also introduced, enabling nodes to evaluate the reliability of their neighbors based on past interactions, thus promoting secure collaboration across the network. A considerable portion of this research tackled well-known threats such as Sybil attacks [14], where a node forges multiple identities; sinkhole attacks [7], which lure traffic by falsely advertising optimal routes; and selective forwarding [8], in which a malicious node drops specific packets to disrupt communication. These threats were addressed by developing detection and mitigation strategies specifically tailored to function under WSN constraints. In this study, we build upon these foundational works by examining several critical areas, including intrusion detection mechanisms for wireless environments, simulation of advanced communication protocols aka. routing protocols using OMNET++, and the implementation of energy harvesting techniques to extend the operational lifetime of sensors, all of which are explored through the lens of the recent literature and experimental validation.

### 2.2. Intrusion Detection System (IDS)

An intrusion detection system (IDS) functions as a vigilant network watchdog, continuously observing traffic to identify and react to potentially harmful or unauthorized behavior, playing a critical role in safeguarding digital infrastructure [15]. Intrusion detection systems (IDSs) have become essential in safeguarding evolving network technologies against rising cyber threats. As these technologies advance, there is a continual emphasis on enhancing IDS capabilities to keep pace with emerging security challenges [16]. Intrusion detection systems (IDSs) typically have two main categories depending on their method of identifying threats: signature-based and anomaly-based. The anomaly-based variant operates by constructing a baseline profile of normal user behavior, flagging any significant deviations as potential threats. This method excels in recognizing novel and previously unseen attacks; however, it tends to produce a higher rate of false positives by occasionally misidentifying legitimate behavior as malicious. Such systems are particularly effective in uncovering network layer threats like sinkhole attacks [17,18]. Conversely, a signature-based IDS relies on predefined patterns derived from known threats to identify intrusions. Its strength lies in maintaining a low false positive rate, as it only triggers alerts when matching a recognized attack signature. However, it lacks the ability to detect new, unfamiliar threats. This method is well suited for identifying established attack types, including jamming attacks at the physical layer. Despite their strengths, IDS implementations are prone to classification errors that may undermine their reliability. A false positive error (FPE) occurs when legitimate activity is wrongly flagged as malicious, while a false negative error (FNE) arises when an actual attack is overlooked and treated as normal behavior [17,18]. Reducing both error types remains as a critical focus in improving IDS accuracy and effectiveness. Trust-based IDSs are also relatively popular. In trust-based intrusion detection systems, the behavior of each node is assessed by examining the variation in specific protocol-level metrics. Since attacks tend to influence particular protocol parameters, any unusual deviation from normal values can be a strong indicator of malicious activity. Nodes continuously observe the actions of their peers and use these observations to calculate a trust score. This trust value is then reported to the base station (BS). If the computed trust level falls below a predefined threshold, the node in question is flagged as suspicious or compromised [19,20]. Reputation-based systems, in contrast, rely on a collaborative approach where trust is established not just through individual observations but through shared evaluations across the network. Each node computes its trustworthiness using a combination of direct trust (based on firsthand interactions) and indirect trust (based on feedback from neighboring nodes) [21,22], enabling more comprehensive and community-driven anomaly detection.

### 2.3. Advanced Routing Protocols

Wireless sensor networks (WSNs) find applications in a wide range of fields, often requiring the deployment of numerous sensor nodes distributed over expansive terrains. In such settings, it is usually impractical for all nodes to communicate directly with one another, making multi-hop communication essential for relaying data. Routing protocols are responsible for establishing and maintaining these communication paths, acting as the backbone for reliable data exchange. The efficiency and suitability of a particular routing method depend heavily on the resource limitations of the sensor nodes, such as energy and processing power, as well as the unique performance requirements of the application in which the network is implemented. In OMNET++, while using frameworks like INET, Castalia, and SimuLTE (discussed in Section 3), several routing protocols can be implemented such as the following:AODV: The AODV (Ad hoc On-Demand Distance Vector) protocol is used in the INET-MANET framework. It has been recognized as a foundational component in the evolution of routing strategies for MANETs. AODV has influenced modern routing developments and is frequently referenced in contemporary research, either for optimizing node configurations [23,24] or for enhancing path discovery mechanisms [25,26,27]. Designed specifically for low-bandwidth scenarios, AODV’s lightweight and adaptive nature makes it highly suitable for simulation and performance testing. As a reactive protocol, it establishes routes only when required, utilizing distance vector routing [28] to dynamically identify the most efficient path between a source and destination. Distance vector routing operates on RPL (Routing Protocol for LLNs) [29], determining both the distance and direction for network links employed in an LLN. Its ability to select the shortest route makes it ideal for networks with minimal traffic. However, under increased channel load, the protocol’s reliance on shortest path routing can lead to decreased performance, packet loss, and eventually network degradation, highlighting a trade-off between simplicity and scalability in congested environments.LEACH: Low-Energy Adaptive Clustering Hierarchy (LEACH) is a cluster-based approach [30], applied in the Castalia framework. The purpose is to use energy evenly, across the entire network. Logically, the network is split into segments, small areas called clusters, and each cluster has a header at its center. The header creates and manages a TDMA [31] (Time Division Multiple Access) schedule to avoid communication collisions between nodes in the region. The cluster head’s role is to collect data from surrounding nodes, aggregating it and forwarding the summarized information to the base station. The LEACH protocol has two main stages: (1) the setup phase and (2) steady phase. First, in the setup phase, each node within a defined sector shares its residual energy along with a randomly generated value with neighboring nodes. The node with the highest energy will become the cluster head. If multiple nodes exhibit equal energy levels, the random number is used to resolve the tie and finalize the CH selection. Once the CHs are determined, the steady-state phase begins. At this stage, data sensing and transmission take place. Regular nodes forward their collected data to their designated CH, which then processes and aggregates these data before transmitting to the base station. This hierarchical structure significantly reduces energy consumption and enhances communication efficiency within the network. In this stage, if the attacker is a general node, it is able to intentionally not comply with the TDMA schedule to cause a collision, such as not transmitting data or transmitting data at the same time as other nodes to interfere with the normal data collection of the header [32].

### 2.4. Energy Harvesting in WSNs

Wireless sensor networks (WSNs) have gained significant traction for their ability to continuously monitor environmental conditions and collect real-time data across diverse settings. Despite their utility, WSNs face several operational challenges that vary with the application context, including issues related to secure communication, efficient data handling, scalability, and resilience. One of the most pressing limitations across all scenarios, however, is the dependence on limited energy sources. Since sensor nodes typically operate on finite battery power, frequent battery replacements or recharging cycles [33,34] become necessary, leading to increased maintenance demands and added environmental impact. To overcome this bottleneck, the adoption of ambient energy harvesting technologies has been proposed, allowing nodes to self-power using sources such as light, vibration, or temperature gradients. Integrating such energy-harvesting capabilities into WSNs supports the ideals of Green IoT by enhancing energy autonomy, reducing waste, and ensuring more sustainable and long-lasting deployments. These types of networks are known as self-sustaining networks. They operate autonomously in wireless communication systems, relying on minimal external energy or human involvement. These networks independently harvest and regulate their energy, utilizing “energy harvesting technologies” like solar, wind, and radio frequency (RF), as illustrated in Figure 2. This approach ensures uninterrupted functionality of devices within the network.

### 2.5. Challenges and Future Work Direction

Although the research work in wireless sensor networks has led to huge improvements in routing protocols and in solving security issues, their exposure to unsecured or hostile environments makes them prime targets for security breaches. The heterogeneous nature of WSNs makes it difficult to imply security mechanisms that may result in failure of achieving security at various layers. WSNs are vulnerable to a range of attacks, such as eavesdropping, node compromise, Sybil attacks, and denial of service. If an attacker gains control of a node, they may extract its stored keys, enabling unauthorized access or disruption of network operations. This makes it crucial for key management mechanisms to be resistant to compromise and resilient against such threats. A cornerstone of securing these networks lies in the efficient generation and handling of cryptographic keys, which are essential for maintaining communication confidentiality, verifying authenticity, and protecting data integrity. As WSNs scale to encompass hundreds or thousands of nodes, managing cryptographic keys becomes exponentially more complex. Distributing, renewing, and revoking keys in a secure and efficient manner is a major challenge, especially as network size increases. Nodes typically operate with strict limitations on processing capability, memory availability, and energy supply. These constraints render traditional cryptographic methods impractical, as they demand resources beyond what sensor nodes can afford. To accommodate these limitations, specialized lightweight encryption techniques and low-overhead key management strategies must be employed. This complexity is further compounded by the dynamic behavior of many WSNs; frequent node failures, mobility, or changes in topology due to environmental factors require continuous adaptation of secure communication channels.

To address these dynamics, some solutions propose generating encryption keys on demand based on real-time network conditions. Although this offers greater flexibility, it also introduces synchronization and distribution difficulties, particularly in large-scale or frequently changing networks. Moreover, securely transmitting keys in such environments is a non-trivial task. If encryption keys are intercepted during distribution, the security of the entire network may be compromised. Ensuring efficient key updates while minimizing energy and communication overhead remains a central challenge.

Given these complexities, there is a clear need for continued research into adaptive, secure, and resource-conscious key management strategies. Advancing these areas will be essential for building robust and future-ready security frameworks in WSNs.

To sum up everything that has been said so far in this literature review, maintaining data security during data transmission or in communication networks is very hard and also vital. There are many technologies emerging and new methods being implemented for the detection of malicious activities. A thorough review is performed on the existing works that use OMNET++ as the simulation platform, as shown in Table 1. The table contains the key features, which approach is used for implementation, and the pros and cons of the related work. There might not be a correct answer for the question of which one is the best solution. The particular reason is that each technique is superior in its own way.

## 3. OMNET++

OMNET++ is a discrete event simulation framework with an object-oriented approach and a modular architecture [43]. It is utilized for simulating traffic in telecommunication networks, modeling communication protocols, and assessing the performance of complex software systems, among various other uses. The installation steps of OMNET++ for Windows, Linux, mac and other OS can be viewed in [44].

### 3.1. Basic Concepts and Model of OMNET++

An OMNET++ model is composed of modules arranged in a hierarchical structure, enabling communication through the exchange of messages. These models are commonly referred to as networks. The highest-level module, known as the system module, encompasses sub-modules that can recursively contain additional sub-modules. This hierarchical nesting can be as deep as needed, allowing the model’s structure to represent the logical organization of the real system accurately. Modules that include sub-modules are identified as compound modules, whereas simple modules form the base layer of the hierarchy and contain the model’s algorithms and logic.

As outlined earlier, modules in OMNET++ exchange messages to communicate. In simulations, these messages might represent packets or frames in a network and are capable of carrying complex data structures. Messages can be sent by other modules or generated within the same module. When a module receives a message it previously dispatched, this is termed a self-message, which is often utilized for implementing timer functionalities. Simple modules are capable of sending messages directly to their targets or routing them through predefined pathways using gates and connections.

Gates in OMNET++ are divided into input and output types. Output gates are used for sending messages outward, while input gates are used to receive incoming messages. Gates are linked by connections, which are essential for forming communication paths. Each connection in the network model is characterized by three essential parameters that influence communication behavior: propagation delay, bit error rate, and data rate. Propagation delay refers to the time interval required for a message to traverse the communication channel and arrive at its target node. The bit error rate indicates the likelihood of individual bits being altered during transmission, allowing the simulation of imperfect or noisy links. Meanwhile, the data rate, defined in bits per second, governs the duration needed to successfully transmit a complete data packet through the connection.

OMNET++ operates with two key programming languages: NED (Network Description) Language and C++. NED is used to define the overall structure and topology of a network and its components. It allows for the creation of reusable component descriptions, promoting modularity within network models. These NED files are text-based and can be created with any text editor, offering a clear and readable depiction of the network’s layout. In contrast, C++ is employed to implement the behavior of individual modules, such as messages and queues. It provides full programming flexibility, supported by the OMNET++ simulation class library. This allows simulation developers to use advanced C++ features, including object-oriented principles like inheritance and polymorphism, as well as design patterns, to extend the simulator’s capabilities.

An OMNET++ simulation consists of the following:NED files (.ned), which outline the module structure, including parameters and connections;C++ source files (.cc and .h) for simple modules;Initialization files (.ini) that set parameter values defined in the NED files.

This modular approach ensures the separation of network topology design from module implementation. A simple example code of a network along with connections and simulation is shown in Figure 3, Figure 4, Figure 5 and Figure 6.

### 3.2. OMNET++ Frameworks

OMNET++ serves as a powerful and flexible simulation platform, supporting a diverse range of frameworks designed for specific domains, including communication networks, vehicular systems, and IoT. Its modular design allows for the development of specialized tools and models, making it a valuable resource for researchers and developers. Below is a list a few omnet++ frameworks:INET:The INET framework is the most feature-rich, enabling the simulation of both wired and wireless networks, covering protocols like TCP/IP, mobility, and routing. Simulation can be performed on both wired and wireless link-layer technologies (e.g., Ethernet and Wi-Fi). It includes network components like hosts, routers, and links. It is useful for general networking research and education. The INET framework is utilized for rapid and real-time analysis of various network scenarios. It enables the simulation of internet services while incorporating features such as malware detection. By implementing the INET framework, we can support simulation models that include both fixed and mobile network configurations.Veins:Veins is employed to study network and traffic scenarios, making it a valuable tool for research across various network-related applications. It is specifically designed for Vehicular Ad Hoc Networks (VANETs) and is commonly used to analyze the performance and behavior of roadside units in different scenarios. For vehicular networks, Veins works in tandem with SUMO to model vehicle-to-vehicle (V2V) and vehicle-to-infrastructure (V2I) communications, facilitating intelligent transportation system research. Additionally, Artery builds on Veins by incorporating ITS-G5 protocols for advanced cooperative intelligent transport systems (C-ITSs).SimuLTE:SimuLTE is designed to simulate the data plane of LTE/LTE-A networks, focusing on the Radio Access Network (RAN) and Evolved Packet Core, including eNodeBs, UEs, and the core network. It supports the simulation of LTE/LTE-A in Frequency Division Duplexing (FDD) mode. It includes realistic channel models and supports MAC operations as well as resource scheduling for both uplink and downlink. It supports QoS mechanisms and mobility scenarios and is commonly used in cellular network research [45].Castalia:A domain-specific framework like Castalia focuses on wireless sensor networks and wireless body area networks. It is utilized to assess platform characteristics specific to various applications. Castalia is tailored for low-power wireless networks and radio models, including IoT, healthcare, and environmental monitoring applications. This simulation framework allows the definition of path loss maps but does not guarantee connectivity between nodes.SUMO:The SUMO framework is utilized to assess the impact of infrastructure and policy changes on vehicular networks. As an open-source simulator, SUMO enables the modeling of traffic systems, including vehicles, public transport, and other modes of transportation. It supports tasks such as visualization, network import, emission analysis, and route optimization, making it a versatile tool for traffic system evaluation. SUMO is highly customizable, allowing users to design custom road networks or import real-world maps, and is often integrated with tools like Veins and OMNET++ for vehicular network research. It is widely used for traffic management, ITS studies, autonomous vehicle development, and sustainability analysis, offering realistic traffic models and scalability for small- to large-scale simulations.

Other frameworks like MiXiM provide detailed modeling of wireless communication at the physical and MAC layers, while PowerTAC and PHOLD address energy market simulations and performance benchmarking of discrete-event systems, respectively. Collectively, these frameworks expand OMNeT++ into a versatile simulation ecosystem for diverse research applications.

## 4. Why OMNET++?

There are many other simulators available, but what is different compared with omnet++? Here is a justification for why OMNET++ is different from other simulators.

OMNET++ differentiates itself from other network simulators through its distinctive design, adaptability, and extensibility. It features a modular, component-driven architecture, enabling users to build simulations by assembling reusable modules to represent complex systems. Unlike more rigid simulators with a monolithic structure, it promotes a flexible and scalable approach, allowing researchers to tailor and expand simulations according to their specific requirements. This modular structure not only makes omnet++ ideal for communication network simulations but also suitable for a wide variety of domains, including traffic management, social networks, and distributed systems.

Unlike many simulators that focus on specific areas, such as NS-3 for network protocols or SUMO for traffic, omnet++ covers a much wider range of simulation domains. It is utilized not only for network modeling but also in areas like vehicular networks (Veins), wireless sensor networks (Castalia), intelligent transportation systems (Artery), and others. This versatility allows OMNET++ to accommodate a variety of research fields, establishing it as a comprehensive, general-purpose simulation platform.

OMNET++ is designed to be user-friendly, offering an integrated graphical interface for creating, running, and visualizing simulations. Its IDE (Integrated Development Environment) allows users to visually build simulations by dragging and dropping modules, making it accessible for newcomers. This is a significant advantage over simulators like NS-3, which lacks a built-in GUI and requires more coding.

OMNET++ also boasts an extensive range of pre-built frameworks and extensions, such as INET, Veins, and Castalia, which simplify the modeling of complex systems. These frameworks enhance omnet++’s capabilities, enabling users to simulate internet communication protocols (TCP/IP and routing), vehicular networks, and low-power wireless sensor networks. While other simulators like NS-3 specialize in certain network types, omnet++ offers a broader array of specialized frameworks, making it ideal for interdisciplinary research.

Scalability is another key strength of omnet++, capable of handling both small- and large-scale simulations with minimal performance loss. Whether simulating simple networks or an entire city’s transportation system, omnet++’s modular structure and efficient simulation engine ensure that complex systems can be accurately modeled. Unlike NS-3, which focuses primarily on large-scale network simulations, omnet++ provides a more adaptable solution for various research needs, including interdisciplinary projects.

As an open-source platform, omnet++ allows users to modify and extend its functionality, making it perfect for researchers looking to experiment with new concepts or integrate emerging technologies. Its open nature is complemented by a vast library of components and modules that are constantly updated by the community, further extending its versatility. OMNET++ uses an event-driven simulation engine that efficiently handles discrete events over time. This approach is especially suited for network and communication simulations, where actions such as packet arrivals and network state changes trigger specific events. The event-driven model abstracts the simulation process, allowing researchers to focus on higher-level design rather than low-level details.

Beyond network simulations, omnet++ supports a wide range of interdisciplinary research fields, such as traffic modeling, IoT, smart grids, and social networks. This broad scope of applications makes omnet++ unique compared to more specialized simulators like NS-3 or SUMO, which focus on specific domains like network protocols or traffic simulation. Finally, omnet++ is backed by extensive documentation, a large and active user community, and a wealth of online resources, including tutorials, forums, and research papers. This comprehensive support system is invaluable for new users and researchers seeking guidance or wishing to integrate different models into their simulations. While other simulators also offer community support, omnet++ is known for its detailed and accessible documentation, making it easier for users to get started and succeed.

Even though OMNET++ is great for modular, visually rich simulations and educational purposes, its reliance on external modules and C++ knowledge can be a barrier. A great understanding of C++ is required. Beginners with less programming experience can find it challenging to get started with. The OMNET++ platform limits users from using built-in models; in most cases, users need to build or adapt existing frameworks. As told in Section 3.2, OMNET++ supports diverse frameworks; this can be a downside for the platform as it depends on external frameworks. For better understanding, a comparison between OMNET++, NS-3, and MATLAB is given in Table 2.

Finally, OMNET++ distinguishes itself from other simulators through its flexible, modular, and extensible architecture, making it suitable for a wide range of applications beyond traditional network simulation. Its graphical user interface, combined with an active community, a rich set of frameworks, and scalability for complex systems, makes OMNET++ a versatile and user-friendly choice for researchers across multiple disciplines. While other simulators like NS-3 or SUMO are highly specialized, OMNET++ provides a broader, more customizable solution for interdisciplinary research.

## 5. Security Attacks in Wireless Networks

### 5.1. DDoS Attack

A Distributed Denial-of-Service (DDoS) attack is a malicious attempt to disrupt the availability of a target by overwhelming it with excessive, unnecessary traffic. The attacker accomplishes this by using a network of compromised devices to flood the victim with pointless packets, making it unable to process legitimate requests. To carry out such an attack, the attacker first builds a botnet, a network of infected hosts that can generate and direct large volumes of malicious traffic. The attacker identifies vulnerable systems, installs malicious software on them to turn them into “zombie” hosts, and configures them to locate and manage additional compromised devices. These zombies exploit further weaknesses in other vulnerable systems, creating an expanding army under the control of master nodes, which are ultimately overseen by the attacker. The DDoS attack is visualized in Figure 7. Zero-day DDoS attacks are a newly emerging form of DDoS attack that leverage unidentified vulnerabilities in systems. Such attacks have increasingly gained popularity among cyber attackers [37,46].

DDoS attacks can be categorized as follows: 1. Attacks based on volume: These attacks focus on saturating the target’s bandwidth, making the website or server unreachable by legitimate users. Common examples include UDP floods, ICMP floods, and spoofed packet floods [47]. 2. Attacks based on protocol: This type targets the server’s resources or intermediate network devices such as firewalls, rather than focusing directly on bandwidth. Examples of protocol-based attacks include SYN floods, Ping of Death, and fragmented packet attacks. 3. Application layer attacks: These attacks overload the application server by sending numerous requests that appear legitimate. Such attacks are becoming more popular because they require fewer resources to overwhelm the target and are more challenging to detect and defend against. Notable examples include Slowloris and SIP INVITE floods [6].

In [48], various traffic scenarios and attacks including DDoS and black hole attacks were simulated. The system architecture is divided into two primary components: the VANET simulation and the security simulation. The VANET simulation involves visualizing vehicle movement and packet transmission using the Objective Modular Network Testbed in C++ using OMNET++ in conjunction with the SUMO traffic simulator. For the security simulation, an external plugin, Snort, is integrated to identify malicious threats. This plugin uses a signature-based verification mechanism to detect potential attacks and notify network administrators. The simulation was carried out across four different environments: urban, highway, semi-urban, and rural. The detection rates achieved in these scenarios were 74.4%, 78.0%, 79.0%, and 77.3%, respectively. These results demonstrate that the Snort plugin was able to detect threats with a satisfactory level of accuracy.

### 5.2. Synchronize (SYN) Flood Attack

Infrastructure layer attacks aim to disrupt networks by taking advantage of weaknesses. They are mainly of two types: protocol-based and volume-based attacks. Methods like amplifying traffic, reflecting requests, and manipulating IP addresses are used to clog the network. Protocol-based attacks, such as SYN floods, drain server resources, while volume-based attacks, like UDP/TCP floods, overwhelm the bandwidth, wasting a lot of network capacity [37,46].

A SYN flood attack exploits a vulnerability in the second phase of the TCP three-way handshake, as illustrated in Figure 8a. This issue arises when a server, after receiving a SYN request, creates a half-open connection by setting up a Transmission Control Block (TCB) to monitor the connection and reserving resources to finalize the handshake. Figure 8b demonstrates how attackers take advantage of this mechanism by sending an excessive number of SYN requests to the server. This forces the server to allocate resources for each incomplete connection, effectively making it unavailable to handle legitimate traffic.

Even if the server reinitializes or releases its resources, the relentless nature and volume of the SYN flood can quickly exhaust the server’s capacity again. The attack’s potency is enhanced by exploiting the time the server waits for an ACK response. The attacker can choose not to respond to the server’s SYN-ACK packets or can use spoofed IP addresses when sending SYN requests, causing the server to direct SYN-ACK responses to non-existent or unresponsive addresses. This prevents legitimate connections from being established and further depletes the server’s resources.

The approach introduced in [49], known as SFaDMT, aims to efficiently identify and filter SYN packets within incoming network traffic. As traffic enters the network, it is initially evaluated by matching its pattern against a database of known signatures. If no direct match is found, the traffic is further analyzed using the SFaDMT mechanism, which compares it with stored signature patterns. This process evaluates the legitimacy of the traffic: if the similarity score exceeds 71%, the traffic is marked as suspicious and access is denied. However, if the similarity is below 69%, the traffic is classified as safe and is granted access to the network. To test the system, simulations were performed in OMNET++, with network sizes varying from 10 to 200 nodes. The simulated environment featured nodes (ranging from 20 to 300) that generated burst traffic to test the system’s ability to handle dynamic detection scenarios. Simulation outcomes showed that SFaDMT significantly outperforms traditional pushback mechanisms, offering a 26% improvement in detecting SYN flood attacks.

### 5.3. Sinkhole Attack

In a sinkhole attack, malicious nodes deceive other nodes by providing false routing information, causing all data to be routed through the compromised node. This can lead to inaccurate responses and the depletion of energy in nearby nodes. Sinkhole attacks may also modify or drop data during transmission. The sinkhole attack is especially detrimental due to the inherent weaknesses of the sensor nodes, including low processing power and limited battery life. The presence of sinkhole attacker nodes can result in messages being dropped, modified, or delayed. This significantly jeopardizes the functioning of WSNs by obstructing timely delivery of information to the base station and disrupting other essential network features [50]. The sinkhole attack is visualized in Figure 9.

The commonly used technique for mitigating sinkhole is a mitigating strategy, which is segregated as three types, distributed monitoring, analysis models, and trust-based models [51]. Traditional security strategies, such as encryption and authentication, are often limited by the resource constraints of WSNs, making them less feasible. Additionally, relying solely on preventive measures is insufficient due to the inherent vulnerabilities of these networks. As a result, intrusion detection systems (IDSs) are essential as a secondary defense mechanism. IDSs work by monitoring network activity to detect and counteract malicious behavior. To boost the performance of IDSs in WSNs, techniques like swarm intelligence and optimization methods are widely used, offering effective solutions to manage the complexity and limitations of WSNs, such as energy constraints. A sinkhole attack is highly unpredictable. In wireless sensor networks, packets are transmitted based on routing metrics defined by specific routing protocols. A compromised node manipulates its routing metrics to deceive neighboring nodes and redirect all their data through itself before reaching the base station. For instance, the method used by a compromised node in a network employing the TinyAODV protocol [52] differs from that in a network using the MintRoute protocol. While MintRoute relies on link quality as the routing metric, TinyAODV uses the number of hops to the base station. Consequently, the techniques for launching a sinkhole attack vary depending on the routing metrics used by the protocol [31,53]. In Table 3, analysis of performance of existing work is shown.

## 6. Conclusions

This paper presents basic concepts and frameworks of OMNET++. Different intrusion attacks in wireless sensor networks are discussed. DDoS attacks, SYN flood attacks, and sinkhole attacks are presented. According to existing studies, many researchers focus on developing ICT-based approaches to detect, identify, and mitigate attacks in wireless sensor networks. Most research has encountered difficulties in tackling security challenges related to the resource constraints and mobility of wireless sensor nodes. Some solutions have focused only on static networks, with a few addressing mobile networks. Very few studies have tested the security systems in actual wireless sensor networks. Moreover, some solutions have shown poor detection rates, high network overhead, and elevated communication costs. Future approaches should aim to reduce network overhead and computational demands, increase detection accuracy, and be validated in real sensor network environments. This would help assess whether the solutions are suitable for the resource limitations of WSNs, such as memory capacity.

## Figures and Tables

**Figure 1 sensors-25-02972-f001:**
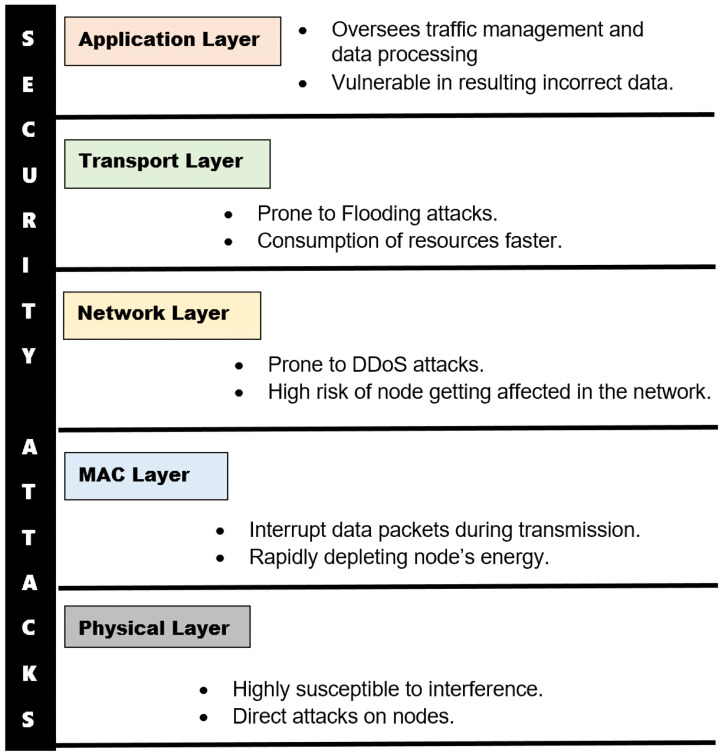
Layer-specific attacks in a WSN.

**Figure 2 sensors-25-02972-f002:**
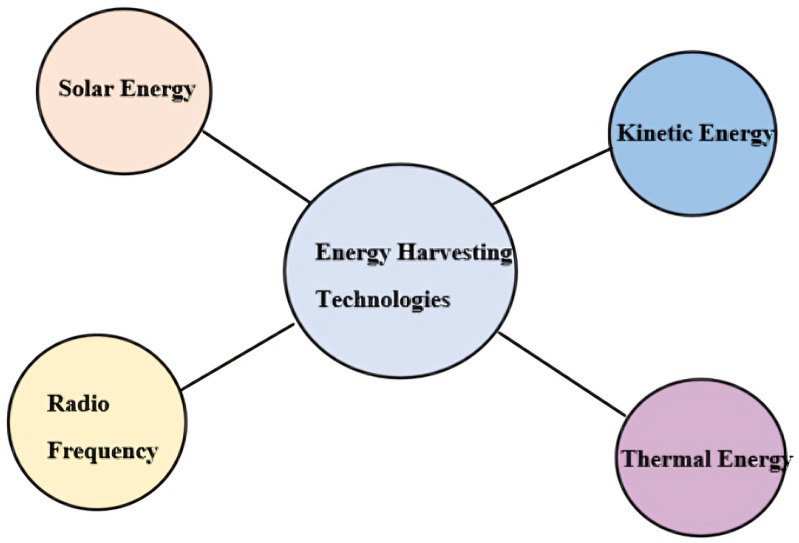
Energy harvesting technologies.

**Figure 3 sensors-25-02972-f003:**
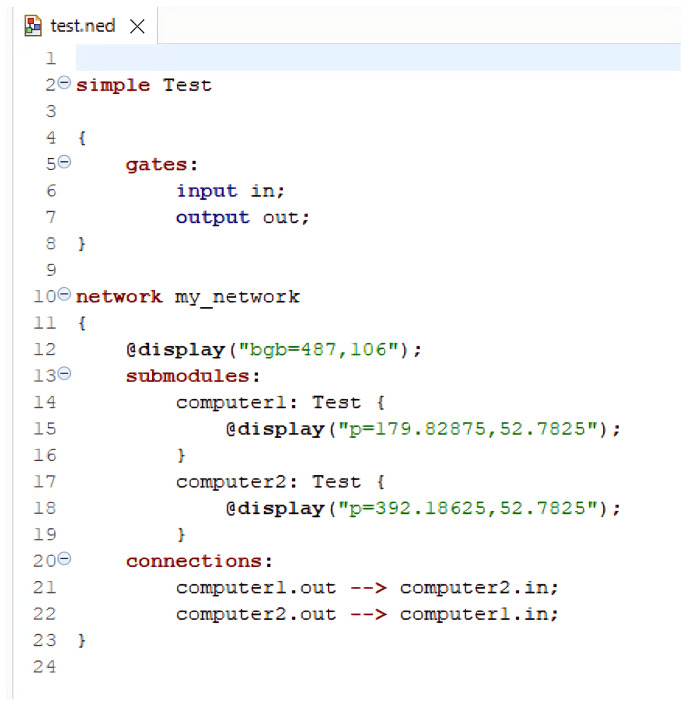
Example for .NED file.

**Figure 4 sensors-25-02972-f004:**
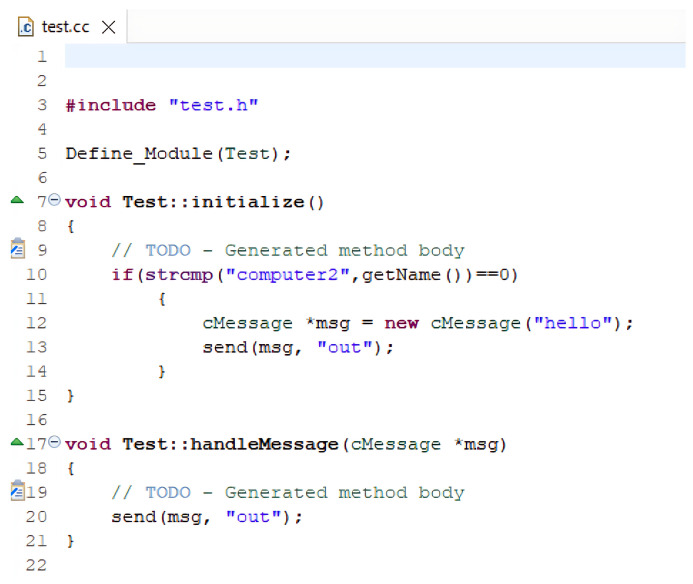
Example for .cc file.

**Figure 5 sensors-25-02972-f005:**
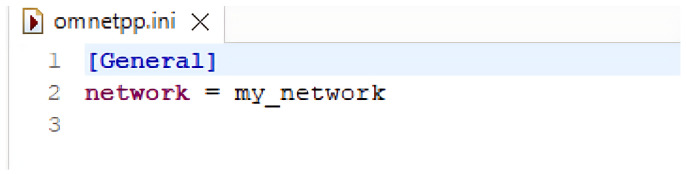
Example for .ini file.

**Figure 6 sensors-25-02972-f006:**
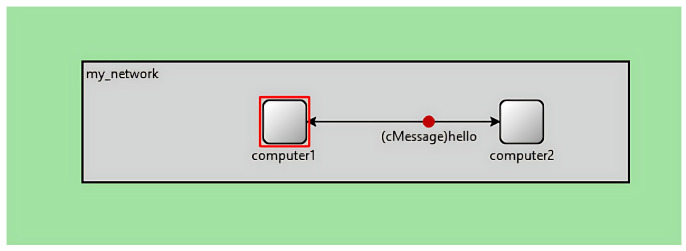
Example for simulation.

**Figure 7 sensors-25-02972-f007:**
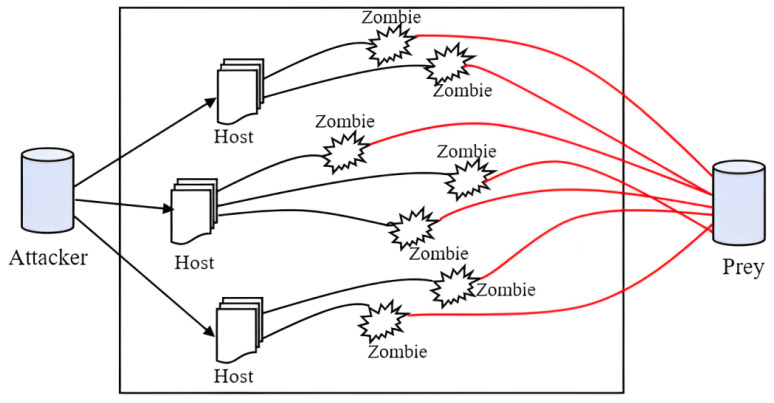
DDoS attack.

**Figure 8 sensors-25-02972-f008:**
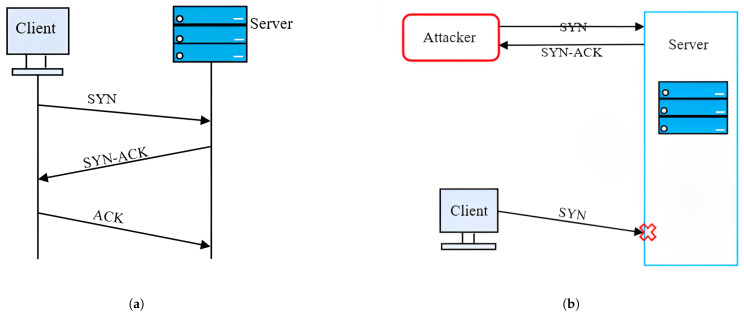
(**a**) TCP three-way handshake; (**b**) SYN flood attack.

**Figure 9 sensors-25-02972-f009:**
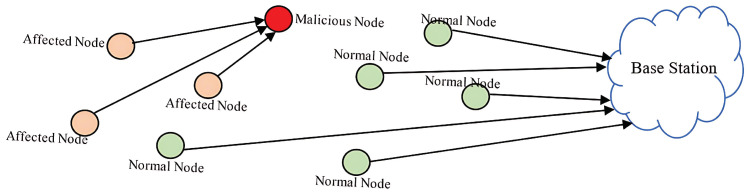
Sinkhole attack.

**Table 1 sensors-25-02972-t001:** Existing work on WSNs using OMNET++.

Related Paper	Key Features	Approach	Pros and Cons
[31] (Nov 2024)	This paper deployed a resource-efficient algorithm for DDoS attacks by combining machine learning and metaheuristic optimization models.	Uses PSO algorithm in the gateway, which will control traffic. They used two rules for allowing normal requests and abnormal requests. The simulation is performed in OMNET++.	The suggested approach offers significant benefits, including precise identification of DDoS attacks, improved system performance, and increased network efficiency. Notably, it achieves a high packet delivery ratio, demonstrating the network’s ability to maintain robust and reliable communication even when subjected to DDoS threats.
[35] (Mar 2023)	In this paper, they proposed a lightweight anomaly detection system for black hole attacks. A data set was developed for analyzing the traffic and studying node behavior.	Using a support vector machine, they classified patterns of attacker nodes and separated the normal-behaving nodes and malicious nodes. The simulation was carried out in OMNET++.	A new data set was generated using a machine learning model and OMNET++. But the limitation of this paper is that the simulation was performed on seven nodes only and there was only one attacker node.
[36] (Mar 2025)	This paper introduces a high-accuracy time synchronization algorithm that integrates the SharkNet protocol with the IEEE 1588 standard to enhance both synchronization precision and overall network performance. The synchronization process works by exchanging timestamp data between parent and child clocks to calculate the time offset.	The model uses a parent clock that transmits messages containing timestamp information to the child clock. These messages are sent at predefined intervals set by the network. When the next scheduled interval is reached, a new round of time synchronization is initiated. OMNET++ is used for the simulation model.	Even though this shows high accuracy and better network performance, if the parent clock has a problem all the child clocks can become inaccurate.
[37] (Dec 2024)	A data set is generated using OMNET++ along with applying deep learning and machine learning algorithms.	Two scenarios with normal data traffic and DDoS attack traffic are created, by generating a big size of packets and transmitting them at high speed. They generate a data set with 512,666 samples along with 16 features.	Using various deep learning and machine learning models, a new data set is created that can be used for research work.
[38] (Mar 2024)	This article introduces a security process based on identifying and verifying each sensor node individually, ensuring that only verified nodes are permitted to exchange data and contribute sensed information within the network.	In the proposed network model, SDAAA employs the base station to verify and grant access to nodes, enabling the data aggregator to reject aggregated data from any node that has not been officially tagged as part of the network.	This approach is novel, maintains data validity, uses multi-attribute authenticity, and is also energy-efficient.
[39] (Aug 2024)	This article proposes an energy-efficient encryption for WSNs that uses hybrid lightweight methods for object detection.	The methodology has two phases: object detection and lightweight encryption. To strengthen security, the model integrates multiple lightweight encryption techniques. It uses a symmetric encryption algorithm to protect key objects, while also applying a pixel scrambling technique that rearranges the entire image’s pixel structure through permutation and shuffling.	This approach improves security by using lightweight encryption and image scrambling to protect data while being efficient on devices with limited resources. However, it may slow down the system, reduce image quality, and could still be vulnerable to advanced attacks. Additionally, it may not work well in all types of systems.
[40] (Oct 2024)	The article proposes a shortest queue length cluster-based routing protocol for a self-sustaining network. This platform supports functions like the initial setup of satellite self-organizing networks and the ongoing maintenance of clusters.	This paper’s simulation is carried out on the satellite self-organizing network platform, concentrating on the performance of packet delay and packet loss rate when routing decisions are made using SQL-CBRP. The results are compared to the Dijkstra algorithm, which minimizes hops, and the GPSR algorithm, which focuses on the shortest path distance.	The advantages of this approach include improved routing performance with reduced packet delay and loss. However, it may face limitations in highly dynamic networks, where frequent changes in topology could affect routing accuracy and stability.
[41] (Jan 2025)	This paper introduces a security positioning technique that can withstand three types of attacks. It identifies attack nodes by analyzing the physical characteristics of each node.	The analysis reveals that the average positioning error of the witch attack algorithm grows rapidly as the number of virtual nodes increases, reaching approximately 80 when four attack nodes are present.	While this method promotes consistency, it may experience slower convergence rates and reduced precision when compared to centralized synchronization approaches.
[42] (Mar 2025)	Proposes a malicious node intrusion detection method for WSNs, which is based on the genetic algorithm optimization of the LEACH hierarchical routing protocol. By optimizing the LEACH protocol with the genetic algorithm, the method incorporates a reputation evaluation mechanism to identify and eliminate malicious nodes.	The genetic algorithm is employed to optimize the LEACH protocol, with the introduction of a hierarchical energy-saving method. This approach focuses on observing and evaluating the behavior of nodes during communication. The study leverages Bayes decision theory, using the beta distribution, to construct a reputation model for WSNs.	The advantages of this approach include improved energy efficiency and enhanced security with a reputation-based node evaluation. However, the method may face limitations in terms of increased computational complexity due to the use of genetic algorithms and the need for continuous monitoring, which could strain network resources.
[10] (Apr 2024)	A trust-based IDS is proposed to incorporate a security mechanism into routing protocols in LLNs.	The methodology employs a distributed and a central approach. A trust-based strategy is used, which calculates trust status values, and based on the threshold the node will be categorized as normal or attacker.	This approach reduces the computation complexity and power consumption issues. But there might be too much work on the root node because of the centralized system.

**Table 2 sensors-25-02972-t002:** Comparing OMNET++ with other simulators.

Features	OMNET++	NS-3	MATLAB
Application Areas	WSNs, IoT, VANETs, MANETs	Internet Protocols, 5G/6G, SDN, IoT, LTE networks	Control systems, robotics, signal processing, power systems, AI modeling
Primary Use	Discrete event network simulation, modular design	Packet-level simulation, protocol evaluation	Numerical computing, system modeling, control design
Programming Language	C++ with NED configuration	C++ and Python	C/C++ and MATLAB
Frameworks	INET, Veins etc.	IPv4/IPv6, LTE, etc.	Extensive Libraries
GUI	It offers a graphical runtime interface like Eclipse-based IDE and host of other tools.	Does not have built-in GUI, depends on external tools like Wireshark, PyViz.	Has its own built-in GUI tool App Designer.
Cost	Open-source and free	Open-source and free	Paid (license required)

**Table 3 sensors-25-02972-t003:** Analysis of performance for existing work.

Ref No.	Method Used	Number of Nodes	Attack Type	Reliability	Throughput	Efficiency	Accuracy
[7] Jun 2024	ABC Optimization Method	150	Sinkhole Attack	Not Given	Not Given	98%	97%
[31] Nov 2024	PSO-ML model	20	DDoS Attack	99.65%	23,446.861 KB	41.651 s (Processing time)	Not Given
[35] Mar 2023	LADS model	7	Blackhole Attack	Not Given	Not Given	Not Given	99%
[38] Mar 2024	SDAAA Method	540	Sybil and Sinkhole Attack	99.5%	444 kbs	98.5%	Not Given
[39] Aug 2024	Energy-Efficient Encryption model	100	Object Detection and Encryption	High	Not Given	89.71 ms	84%

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
