# Peer review of "Security in Wireless Sensor Networks Using OMNET++: Literature Review"

_sensors, 2025, doi:10.3390/s25102972_

Round 1
Reviewer 1 Report
Comments and Suggestions for Authors
This paper provides a comprehensive overview of the security challenges in Wireless Sensor Networks (WSNs). It explores the various application areas of WSNs, such as cybersecurity and data protection, and illustrates the fundamental techniques and methodologies employed in WSN simulations using OMNET++. However, upon careful reading, I still have the following concerns.
- Although the paper offers a thorough summary of existing research, it falls short in identifying research gaps and proposing innovative research directions. It does not effectively highlight how to achieve breakthroughs based on the current state of research.
- Suggest modifying all the images in the article to vector or high-definition images to improve the readability for readers.
- The logical coherence of some sections is lacking. For instance, when introducing the OMNET++ frameworks, the descriptions of each framework appear relatively independent and do not integrate deeply with the overall research problem.
- In Section 4, while the advantages of OMNET++ are emphasized, its potential disadvantages or limitations are not discussed. It is recommended that the paper include appropriate mention of these shortcomings to provide readers with a more objective reference for future research.
- In Section 5, the impacts of various attacks are described mostly in qualitative terms, without specific data or experimental results to quantify their harm levels. This makes it difficult for readers to fully understand the severity of these attacks.
- While the paper focuses on the application of OMNET++ in the security research of WSNs, it barely mentions how to use OMNET++ to simulate attack scenarios and what defensive insights can be gained from these simulations.
- It is recommended that the paper standardize the reference format and citation order to enhance its readability and academic rigor.
Author Response
Note: I edited in LaTEX format, since the references were appearing as text.
The response is given in word file attached.

Reviewer 2 Report
Comments and Suggestions for Authors
- The topic is relevant: security in Wireless Sensor Networks (WSNs) is critical today.
- You highlight the importance of simulation environments (like OMNeT++).
- You mention frameworks, protocols, and security attacks, which shows good technical coverage.
However,
- No future research direction mentioned
- there are no results or findings for the current related works and future research direction.
- 'This section presents a comprehensive review of prior studies and associated research within the field.' At least you have to mention the years of these studies e.g. from 2023-2025.
- there are errors in references in the paper e.g., \cite{47} there are no compilation results.
- the AODV: The AODV (.... and LEACH: Low-Energy ..... should appear as bullet-form points.
- in Table 1.- the year of publication should be inserted.
- Some recent statistics should be inserted. The
- Omnet++ architecture/framework should be highlighted.
- Although this work appears to be primarily a literature review, it is important to highlight the main findings, key challenges identified, and recommendations for future research.
Author Response
Note: I edited in LaTEX format, since the references were appearing as text.
Comment1: [No future research direction mentioned]
Response1:[I agree that paper does not includes future work, so I added a section 2.5 (page 7) which says about the Challenges and Future Work Direction.]
Comment2: [there are no results or findings for the current related works and future research direction]
Response2:[By deeply considering your comment we added a part where we reviewed a paper and explained how OMNET++ was used along with results for that attack. (section 5, page no. 17,1819).]
Comment3:[This section presents a comprehensive review of prior studies and associated research within the field.' At least you have to mention the years of these studies e.g. from 2023-2025. ]
Response3:[Year of publication added in the table]
Comment4:[there are errors in references in the paper e.g., \cite{47} there are no compilation results]
Response4:[That was because i wrote the paper in Word Template,now i changed to LaTEX form so the references appear as link instead of text.]
Comment5:[the AODV: The AODV (.... and LEACH: Low-Energy ..... should appear as bullet-form points.]
Response5:[Bullet points added for the suggested part.]
Comment6:[in Table 1.- the year of publication should be inserted.]
Response6:[As suggested Publication year added.]
Comment7:[Some recent statistics should be inserted.]
Response7:[We added a table of performance analysis for recent papers (section 5, page no. 19)]
Comment8:[The Omnet++ architecture/framework should be highlighted. Although this work appears to be primarily a literature review, it is important to highlight the main findings, key challenges identified, and recommendations for future research.]
Response8:[For reader's understanding we added images showing code for .NED and C++ files. (page no. 12,13). and "challenges and future work" added in page no. 7]
Round 2
Reviewer 1 Report
Comments and Suggestions for Authors
I think that the authors have answered all my questions.
Reviewer 2 Report
Comments and Suggestions for Authors
The authors addressed my concerns, I recommend accepting the paper.
Comments on the Quality of English LanguageThe English could be improved to more clearly express the research